# Conversational Agents for Energy Awareness and Efficiency: A Survey

**Manuela Sanguinetti \*** and **Maurizio Atzori**

DMI—Department of Mathematics and Computer Science, University of Cagliari, 09124 Cagliari, Italy; atzori@unica.it
\* Correspondence: manuela.sanguinetti@unica.it

**Abstract:** The need to reduce greenhouse gas emissions and promote energy efficiency is crucial to achieve the energy transition and sustainable development goals. The availability of tools that provide clear information on energy consumption plays a key role in this transition, enabling users to monitor, manage, and optimize their energy use. This process, commonly referred to as energy feedback or eco-feedback, involves delivering information regarding energy usage and potentially suggesting more sustainable practices. Within the range of available tools, conversational agents can represent a valuable channel to receive detailed information about energy consumption and tailored advice for improving energy efficiency. The aim of this article is thus to explore the application of conversational agents, focusing on eco-feedback, as these tools are primarily devised to foster user awareness of energy usage and enhance more participatory conservation strategies. To this end, we conducted a keyword-based search of major scientific article databases, applying strict criteria to select relevant studies. The results of the collection showed that there is a very diverse landscape with respect to this topic. The surveyed works exhibit a high versatility in feedback goals. Furthermore, while predominantly applied domestically, they also show potential in commercial and industrial settings. Implementation choices also vary to a great extent, while evaluation practices lack a systematic approach and highlight the need for greater consistency. In light of these remarks, we also outline possible future extensions of this type of application, exploring in particular the emerging challenges associated with the increased use of renewable sources and the rise of local decentralized energy communities.

**Keywords:** conversational agents; eco-feedback; energy informatics

## 1. Introduction

Significant emphasis is currently placed on energy efficiency and its crucial role in accelerating the energy transition, in achieving the sustainable development goals (SDGs) promoted by the United Nations (https://sdgs.un.org/, accessed on 30 November 2023), and in the implementation of the NetZero Roadmap [1]. Although drastic measures must be taken at the national and supranational levels to achieve these goals, the advancement of technology has made it possible for individuals to actively contribute towards these goals, using tools accessible to all. For instance, the advent of smart devices, designed to interact with one another and with users, offers a wide range of possibilities for monitoring and regulating energy use, including in household settings. The increasingly widespread connection of home appliances, heating systems, lighting, and other devices to the smart grid enables not only remote control but also the collection of detailed consumption data. This increased accessibility of resources for better energy management can not only be financially convenient but also reveals a potentially crucial role in reducing environmental impact by promoting more sustainable behaviors. Nevertheless, research has shown that the adoption of such systems alone, without adequate support in reporting energy data and conveying the impact of such devices on consumption, does not generate significant results

in terms of adoption of virtuous behaviors by the end users [2,3]. In this context, therefore, the crucial importance emerges of using effective feedback techniques, or eco-feedback (see Section 2.1), to improve users' understanding of the goals achieved, encouraging more responsible and efficiency oriented behaviors. Eco-feedback mechanisms give consumers a detailed view of their energy consumption patterns. The main objective is precisely to increase awareness about how their use of appliances and devices impacts energy expenses [4]. Techniques based on natural language processing and generation, as well as conversational interfaces, fit into this context as potential eco-feedback processing tools that are meaningful to the user for efficient resource conservation. This potential is especially expressed in the possibility of integrating the provision of clear messages, and useful and personalized advice, with a level of human-like interaction achieved through dialogue, thus fostering greater acceptance of these tools as an aid to consumption efficiency.

Conversational agents have been used extensively in several related fields, with the aim of promoting sustainability and enhancing users' commitment to adopt more eco-friendly habits; for instance, different examples of chatbots for the circular economy are reviewed in Zota et al. [5], while other research projects have focused on the use of these tools to foster sustainable mobility beliefs [6], to increase environmentally conscious behaviors among the employees of an office [7], or to reduce food waste in a household [8].

The goal of this survey is to provide an extensive overview of the state-of-the-art regarding conversational agents in the energy domain, with a focus on the use of tools that provide energy-related feedback through text or voice-based interactions. A large number of works can be found on visualization or gamification techniques devised for the delivery of eco-feedback (see Section 2.1); likewise, countless articles have attempted to provide up-to-date reviews of conversational agents [9–13], and have even applied them to specific sub-fields, such as health [14], business [15], and virtual reality and Internet of Things [16], to name a few. To the best of our knowledge, however, the current article represents the first literature review on the use of conversational agents specifically to convey energy feedback. Our intent is to highlight the diversity of contributions and viewpoints involved in this domain, as well as to provide a clear organization of the methods and techniques found in the literature. This approach aims not only to describe what has been done so far, but also to serve as a starting point for future studies devoted to this topic, highlighting the potential and limitations of conversational agents in this context, as well as possible room for improvement and expansion. We thus aim to help create a solid foundation for future developments, pinpointing the challenges to be addressed and suggesting possible research directions to expand the understanding and effectiveness of such agents in the energy context.

In line with these goals and motivations, the present paper is organized so as to provide a preliminary background on the notions of energy feedback and conversational agents, also mentioning specific projects that lie at the intersection of these two domains (Section 2). It will thus describe the protocol we adopted to conduct the survey (Section 3) and will briefly outline the works identified through the search (Section 4). A discussion (Section 5) will then provide a more detailed account of the reviewed papers, while some closing remarks and suggestions (Section 6) will complement the discussion, underscoring the current limitations of this type of research application and proposing potential future directions.

## 2. Background

### 2.1. Energy Feedback

Research on energy feedback is rooted in the field of environmental psychology [17]. Nonetheless, despite the abundance of literature on energy feedback, especially from the past decade, an agreed-upon operational definition of energy feedback is still lacking [18]. We acknowledge that the body of work devoted to the study of energy feedback is extremely vast and complex, and a broad survey of the literature on this subject would be far beyond the scope of this survey. The purpose of this section is therefore to provide

purely introductory notions of what is meant by this concept, referring the reader to the appropriate references for further inquiry.

In an attempt to encompass and clarify the different perspectives offered by past definitions, Chalal et al. [19] (p. 381) characterize this concept as any "information about actual energy use that is collected in some way and provided back to the energy consumer". In the present work, we further extend this notion (sometimes also denoted with the expression eco-feedback; see, e.g., [19–22]), taking into account its ultimate goal, i.e., that of raising awareness among the end users about their consumption habits and patterns, and eventually promoting more sustainable behaviors by providing meaningful insights from consumption data [20]. Hence, energy feedback is used here as an umbrella term to refer to a wider range of energy-related types of information, which may comprise more or less detailed reports on energy use and consumption, but also energy-saving advice (either based on general recommendations or customized) or actionable instructions.

As pointed out in De Ruyck et al. [23], energy feedback literature revolves around two main research questions: on the one hand, efforts are driven towards exploring the many possible ways to represent such feedback; on the other, special attention is also drawn to the most effective strategies to induce an actual behavior change in users through the eco-feedback. A large number of feedback mechanisms have been proposed over the years, but most of the research on this domain has been tackled by primarily focusing on visualization techniques [19], i.e., resorting to graphical solutions such as charts, dashboards, or infographics that make intuitive and immediate sense of consumption data. In this regard, design efforts have been typically directed towards exploring different ways of visualizing energy information, including historical, real-time, and forecast data. In addition, these visualizations take shape through a variety of devices, such as smart phones, tablets [24], and smart watches [25], but also through web interfaces or via SMS/email notifications [26]. In addition to possible visualization techniques, other studies propose methods for changing consumption patterns, thus exploring eco-feedback mechanisms from a behavioral perspective. The provision of action-oriented suggestions has emerged among the prevalent strategies, also putting special emphasis on the role of personalization [21,27]. The latter in particular can be pivotal in increasing users' energy awareness, and in encouraging energy conservation habits. Gamification as well stands out as a commonly used tool in this area [28], where resorting to challenges or comparisons help the users' in setting effective and achievable goals.

A recent position paper [29] argues instead for the relevance of using conversational agents to provide energy feedback, emphasizing the human-like nature of such tools as a particularly effective means of persuasion. The central argument of this perspective justifies the adoption of conversational agents, not only as channels of automatic responses but also as entities that can emulate human traits, adding a more intuitive and engaging level of interaction. The present survey aims to delve into the body of work that embraces this vision. For this purpose, we provide in the next section a brief introduction to conversational agents and their main characteristics.

### 2.2. Conversational Agents

The need to simulate human interactions with the help of automatic tools has been extensively explored since the seminal work by Joseph Weizenbaum in the 1960s [30], and it has gained an even greater popularity more recently due to the fast-paced advances in artificial intelligence, in Natural Language Processing (NLP) in particular. The user interfaces that involve interactions through natural language are called conversational agents, or dialogue systems.

These agents can be categorized into various sub-types, based on the purpose for which they are developed and the modes of communication adopted for the interaction. In the former case, we can distinguish between chat-oriented and task- or goal-oriented dialogue systems. Chat-oriented, or open-domain conversational agents, are primarily intended for entertainment purposes and are thus designed to mimic the unstructured

nature of human–human interactions [31]. As a result, they typically involve multi-turn dialogues that do not adhere to a pre-defined plan. Conversely, task-oriented dialogue agents are precisely designed to fulfill specific tasks, such as making a reservation, obtaining detailed information on a particular topic, or providing assistance. They usually engage in short interactions primarily aimed at gathering information from the user that is relevant for completing the task [32]. Due to the fact that they are designed for well-defined tasks or activities, these agents are usually domain-specific. Therefore, their ability to understand and respond is limited to the context of the assigned task. Furthermore, to successfully achieve their goals, they often need external knowledge bases that can be used to provide the information requested by the user or even to perform operations on the data in order to complete the assigned task. The type of conversational agents covered by the present study are generally aimed at performing a given task, be it providing feedback or advice, or taking specific actions on a given device, and therefore fall within the category of task-oriented agents.

As far as the mode of communication is concerned, such agents can allow either text- or voice-based interactions. In the latter case, we often refer to this type of dialogue agents with the expression virtual assistants, while the former category is usually labeled with the term chatbots. Other types of dialogue system include embodied conversational agents, designed to incorporate visual elements, such as a virtual avatar, to enable a richer interaction with the help of hand gestures or facial expressions. For a more detailed taxonomy of conversational agents, we refer the reader to Allouch et al. [11].

On a higher-level, the common architecture of a conversational agent, and more in particular of a task-oriented agent, typically follows a modular scheme consisting of three main components (also shown in Figure 1), which we briefly describe below:

- The Natural Language Understanding (NLU) module is used to classify the type of user request (also defined as intent) and identify possible additional information useful to process that request (the so-called slots). Consequently, the NLU module typically manages various tasks, with intent detection and slot filling being the most prevalent. Intent detection specifically involves identifying the primary goal or task within the user's utterance that the agent should address. On the other hand, slot filling entails identifying the text span expressing the parameters required to fulfill the requested action. While the former task is commonly treated as a supervised classification task, the latter is typically approached as a sequence labeling problem, often employing recurrent neural networks. Given the inherent connection between the expected slot labels based on the intent, and vice versa, joint approaches are also frequently employed. These methods range from conditional random fields to more recent transformer architectures [33].
- The Dialogue Management (DM) module controls the conversation flow, triggering the proper action or response based on the identified intent and conversational context, and handling missing information in the user's message; these tasks are managed by two core components, i.e., the Dialogue State Tracker (DST) and the Dialogue Policy (DP) module. The Dialogue State tracker keeps track of the current state of the dialogue, while the Dialogue Policy determines the system's next actions or responses, including assessing whether clarification or follow-up questions are required due to unclear or missing information in the user utterance [31]. In a recent survey by Brabra et al. [34], three main approaches have been identified as common practice while handling both DST and DP tasks: (1) hand-crafted approaches that resort, e.g., to specific rule definition languages such as AIML (http://www.aiml.foundation/, accessed on 28 December 2023) or to finite state machines; (2) data-driven methods (focusing on reinforcement learning in particular); (3) hybrid approaches.
- The Natural Language Generation (NLG) module is precisely adopted to generate the agent's response, resorting either to pre-defined templates or to symbolic or neural-based techniques. Template-based approaches consist of designing pre-written texts that may also be filled with dynamic and contextually relevant information. Traditional

NLG methods, instead, have long adopted the standard pipeline architecture defined by Reiter and Dale [35], comprising three sequential stages for document planning, sentence planning, and surface realization. However, the evolution of neural end-to-end techniques has helped make them widespread in the NLG field as well [13].

For speech-enabled virtual assistants, an automatic speech recognition module is also needed, to convert into text the audio input from the user, while a text-to-speech module is finally used to output the assistant's response starting from the generated text. Furthermore, the dialogue manager can also communicate with external services or databases in order to either directly retrieve the proper answer, or to call auxiliary modules in charge, for instance, of aggregating the data obtained from the external sources, or devoted to specific computing tasks (see the "Data processing modules" box in Figure 1). This is especially true in the specific case of conversational agents aimed at providing energy feedback, as defined above. Useful data sources or services can be represented in this context by a number of integrated components, such as consumption databases, which in turn can be populated by gathering data from sensors and smart meters (installed to measure disaggregated energy usage). Additional sources and services can be the technical specifications of given appliances or public service APIs, e.g., to retrieve data on weather forecasts or energy tariffs. Finally, the integration of smart devices can be equally important, in that they enable a remote control of a given appliance, changing its operation state or settings. This control can be exercised directly by the user via the conversational interface (for example, by requesting to switch on or off the device through text or voice commands), but also autonomously by the agent itself as a result of previous processing steps, devoted, for example, to determining the optimal time to use the given device or its optimal settings, also based on a specific user's comfort needs.

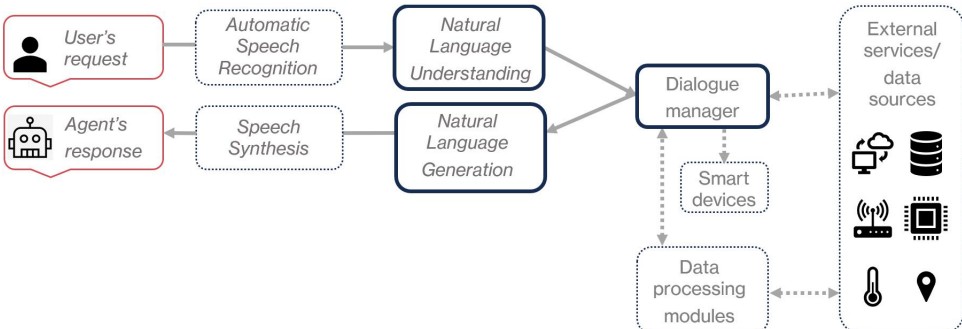

**Figure 1.** High-level architecture of a task-oriented conversational agent in the energy domain. Thick arrows and boxes represent the minimal components typically included in a conversational agent (i.e., Natural Language Understanding, Dialogue Management, and Natural Language Generation), while the dotted ones are used instead to represent additional components that may change depending on the different goals of the agent and the type of service or feedback it is intended to offer. For the sake of simplicity, the box representing the external modules and data sources, as well as the one for data processing, have been conflated in the figure as single blocks, but it is worth pointing out that actual implementations may involve different (and possibly separate) layers that communicate with one another.

As an alternative to the traditional modular pipeline, end-to-end architectures based on deep learning have also emerged recently. These architectures aim to overcome the main challenges encountered in modular frameworks, such as error propagation from one module to another [13]. Despite this, the dialogue systems examined in this overview mainly follow the first-mentioned architecture, motivating our focus on this framework in this section.

Nowadays, chatbots and virtual assistants are widely adopted tools that can be used to perform the more diverse tasks. Commercial products such as Amazon's Alexa or Google Assistant are very common applications that may assist lay-users in their everyday

activities or information needs, but many other domain-specific tasks can also be performed through human-like interactions with these agents, such as health assistance [36], teaching support [37], or customer-care operations [38], to name a few. Concerning the latter, for example, text-based conversational agents are commonly used by companies to manage routine tasks, such as responding to frequently asked questions (FAQs), or to handle complaints or requests for assistance concerning any customer–company transactions.

In conjunction with this diverse range of uses and practical applications of conversational agents, there is also a wide selection of products and platforms that support the development of such systems. This landscape is constantly evolving and is strongly influenced by the major advances made, particularly in the area of NLP and generative AI. Some of the most-popular frameworks include DialogFlow (https://cloud.google.com/dialogflow/, accessed on 30 November 2023), IBM Watson (https://www.ibm.com/products/watsonx-assistant, accessed on 30 November 2023), Amazon Lex (https://aws.amazon.com/lex/, accessed on 30 November 2023), or RASA Open Source (https://opensource.rasa.com/, accessed on 30 November 2023). These platforms feature a comprehensive toolset catering to diverse use case scenarios, though still adhering to a comparable system architecture that is in line with the one described above and shown in Figure 1 [39].

*2.3. Communicating Energy Issues Using Dialogue Agents*

Among others, the use of conversational techniques to provide energy-related feedback has recently emerged as a topic of interest, both in academia and in the private sector. This interest has prompted the development of projects dedicated to exploring the potential of such approaches. One example is GreenLight (https://digitaleducationhack.com/en/solutions/green-light, accessed on 30 November 2023), a chatbot developed within the context of a coding challenge held in 2019 and funded by the EU Commission, i.e., the DigiEduHack (https://digitaleducationhack.com/en/, accessed on 30 November 2023). Its main goal is to provide recommendations on sustainable transport.

Another particularly relevant example is EcoBot (http://eco-bot.eu/, accessed on 30 November 2023), a project funded by the EU Horizon 2020 program. EcoBot's main focus is on creating an advanced chatbot that offers highly personalized feedback. This is achieved through the analysis and processing of a large amount of data from different sources, in particular smart meters, third-party applications (especially for weather forecasting), and custom behavioral models that take users' attitudes towards energy saving into consideration. Three pilot studies have been implemented in this project, two in residential buildings and one for commercial buildings and energy managers, to test the chatbot's capabilities in different scenarios.

In the next sections, we will outline the research contributions identified for this survey, starting with the search and selection protocol defined for our analysis.

**3. Survey Protocol**

The general objective of this survey is to provide an overview of the studies in the literature that pertain to the use of conversational agents to provide energy feedback. In light of these premises, we thus intend to investigate four specific aspects, represented by the research questions formulated below:

- RQ1: What is the primary focus of the works found in the literature dealing with conversational agents for energy feedback?
- RQ2: What are the main goals (in terms of type of feedback and agent's functionalities) and use scenarios for this type of conversational interface?
- RQ3: What are the main approaches adopted for their development?
- RQ4: In the existing body of work, what form of evaluation is carried out on conversational agents for energy feedback? What are the common practices (if any) and evaluation criteria?

Based on these research objectives, we conducted a comprehensive search of all the material available in the literature on the topic of conversational agents in the energy field,

inspired by the recommendations established in the PRISMA 2020 statement for systematic reviews [40]. We thus defined the pool of online sources to be consulted for the research of the material, we established inclusion and exclusion criteria that were in line with the general research objective of this review, and finally analyzed the collected material after an iterative selection process. Below we define this process in more detail, starting with the definition of the selection criteria and then describing the search procedure.

### 3.1. Inclusion and Exclusion Criteria

Our selection methodology focused on the identification of articles that investigate the topic of conversational agents developed with a particular emphasis on energy awareness and efficiency. Within this perimeter, we focused not only on the description of these agents but also on other related aspects, with the aim of understanding how such agents can be used to convey practical advice on energy savings and provide meaningful feedback on energy consumption.

User-centered design, accurate understanding of user's messages and provision of appropriate responses, carefully considering the context of the conversation, or responding to requests using information from knowledge bases or external sources all represent only part of the complex dynamics involved in the development of a conversational agent and show how this topic lies at the intersection of multiple fields. This is especially true in the specific context of this study. Namely, due to the language-oriented distinguishing feature of this type of user interface, Natural Language Processing may play a central role in the development of such systems, but embedded and Internet-of-Things (IoT) devices serve as critical components as well (see Section 2.2), in that they can provide real-time data, monitoring, and collecting information on energy use in a household or commercial/industrial building, even at the level of single appliances. In addition, contributions from the area of Human–Computer Interaction (HCI) can provide valuable insights. We have seen in Section 2.1, for example, how the study of eco-feedback has been widely addressed in this field in order to identify effective mechanisms and strategies to communicate energy issues and highlight the main strengths and limitations of such feedback, as well as its actual long-term impact in terms of real changes in user habits. Likewise, user-centered design—through activities such as interviews, focus groups, prototyping, and usability testing—offers a diverse range of approaches to pinpoint potential areas of intervention, along with the needs, abilities, and values of the intended target users, and thus to improve the quality of interaction in terms of user expectations. In light of the above, the inclusion criteria adopted in this study were deliberately left open to research contributions from any research field, as long as they pertained to the use of conversational agents to communicate energy issues. Furthermore, and complementary to this, we also included some works identified through our systematic search that did not focus specifically on conversational agents as a whole, but rather on the development of systems aimed at handling at least one of the main tasks typically involved within a dialogue system (also summarized in Figure 1). One such example is the generation of textual feedback from structured data, such as that from smart meters or other consumption monitoring tools. Although such systems have been developed as standalone contributions, they can potentially be integrated within a dialogue system and therefore represent a valuable and relevant contribution to this overview.

As regards the exclusion criteria, they were specifically conceived to ensure the consistency and specificity of our research focus. As a result, artifacts that rely primarily on visual and, more generally, non-language-based feedback have been excluded, as this deviates from our natural-language-oriented investigation. In line with this principle, we thus also decided to exclude IoT artifacts that do not involve user interaction through natural language, or the whole body of work that deals with the use of conversational agents for smart homes but do not have energy monitoring and feedback among their main goals or functionalities. As a matter of fact, although the availability of home automation systems can ultimately contribute to a greater energy efficiency, we believe that the use of chatbots

or voice assistants for the mere automation of household appliances (i.e., offering functions for switching devices and appliances on or off) does not fully reflect the specific orientation of our research towards the promotion of energy awareness and efficiency through meaningful linguistic interactions. Finally, we did not consider as eligible contributions for the review the following categories: publications that have not been peer reviewed, such as pre-prints or Master's theses, publications that were not fully accessible, and papers that do not introduce an original contribution (such as position papers). In addition, we limited our analysis to publications written in English, although some of them (as shown in Section 5.3) may regard systems managing non-English conversations.

### 3.2. Information Sources and Search Protocol

As is customary in this type of research, some of the main databases and online search engines for scientific articles have been selected as primary data sources, namely ACM Digital Library, IEEE Xplore Science Direct, and Google Scholar. The research was carried out by querying these sources using a combination of the following keywords: ("chatbot" OR "conversational agent" OR "dialogue system" OR "natural language interface" OR "virtual assistant") AND ("energy feedback" OR "energy consumption" OR "energy efficiency" OR "smart energy"). The search was carried out manually via the corresponding web interfaces, without resorting to APIs or automatized scripts. For none of the sources used for the search was a time constraint set on the date of publication, and in cases where the results provided by the query were of a large number (this was the case in particular for the ACM Digital Library and Google Scholar), only the results that could be consulted up to the fifth page (limit conventionally established after some preliminary tests) were considered for the selection, assuming that increasingly less relevant results would have been found after that limit. With this initial selection, we found a total of 508 records (250 in ACM, 22 in IEEE Xplore, 136 in Science Direct, and 100 in Google Scholar). The search was last updated in late October 2023.

Starting from the initial selection phase, we manually scanned this preliminary collection, only retaining for further inspection the articles we deemed potentially relevant to this survey. This filtering process generated an initial set of 47 items, subject to further assessment. The items were saved as records in a spreadsheet, with columns including the title, the source the paper was retrieved from, its URL, and whether it could be eligible for full screening on the basis of both the title and any keywords defined by the respective authors, along with an additional column with open notes explaining why a given paper should be excluded. During the analysis of this collection, cases of duplicates (i.e., the same article found in multiple databases) and articles that did not meet the peer review criteria or could not be considered as proper scientific contributions (e.g., dissertations) emerged. A preliminary evaluation of the abstracts of the remaining articles led to the exclusion of 12 of them, considered non-relevant as per the selection criteria outlined above, thus refining the final collection to 23 articles.

Finally, the last step involved a full screening of the articles in the selection and their final assessment with respect to the relevance criteria established for this study. At the end of this validation phase, 12 articles were considered fully relevant and included in the review. The spreadsheet created in the previous step was then extended to incorporate additional columns aimed at capturing specific aspects (discussed in Section 5) that we intended to identify within the papers to address our research questions.

In order to ensure that potentially relevant papers were not overlooked in the aforementioned steps, additional research was conducted to assess possible contributions outside the main systematic search. Three contributions, already known to the authors of this manuscript, were included in the selection process. In addition, the works cited as related work in the 12 articles selected in the last phase of the systematic procedure were fully screened. Among these, excluding those already found in the systematic search, five articles were further analyzed. Out of these eight contributions found through non-systematic search, three were finally assessed as relevant and included in the review. This

multi-source and iterative search protocol is also summarized in Figure 2. In the next sections, we will discuss the results obtained after this process.

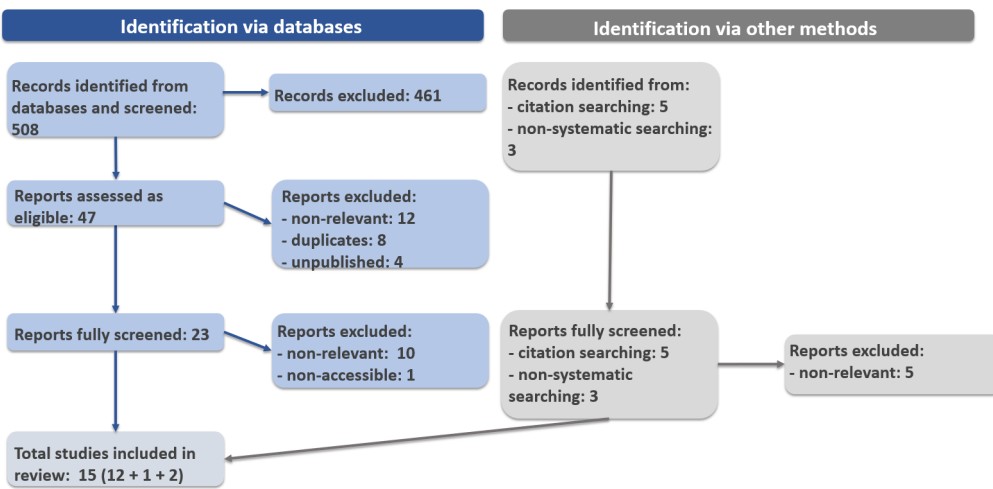

**Figure 2.** Survey protocol summarized with a PRISMA flow chart (adapted from [40]).

## 4. Overview of the Results

The collection used for this review consists of 15 contributions, 5 of which have been published in journals, and the remainder in academic conferences. These works have been published since 2015 but have been distributed more widely in 2018 and in the last three years, as also shown in Figure 3. Below, we briefly outline the contributions we included in this review, also highlighting their primary strengths and weaknesses, while in the next section we will discuss more in-depth their main features in light of the research questions defined in Section 3.

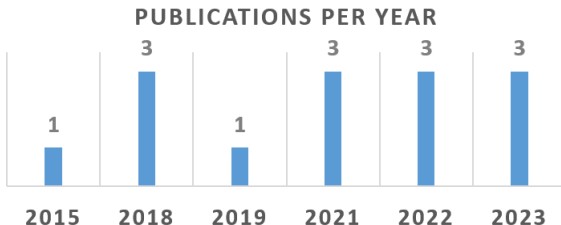

**Figure 3.** Distribution of the reviewed contributions per publication year.

### Alexiadis et al. [41]

The authors develop a personal assistant that integrates into the conversational framework a multi-intent NLU component, a rule-based dialogue manager, and a planning module for the management of household appliances (mainly lights and HVACs) according to a Demand Response paradigm. The proof-of-concept was tested in a real-life scenario within the nZEB Smart Home project, carried out at the Center for Research & Technology, Hellas (CERTH) (https://smarthome.iti.gr/, accessed on 30 November 2023); however, the feedback capabilities of the agent are quite limited and need to be extended.

### Trivino and Sanchez-Valdes [42]

The authors create a data-to-text generation system that generates textual advice on household consumption. The system is based on a model called GLMP (Granular Linguistic Model of Phenomena), for generating personalized text from numerical data. The model is designed to map quantities expressing energy consumption at different degrees of granularity into linguistic expressions. These converted expressions are then included in pre-defined text templates that provide custom advice on how to improve the daily energy

consumption behavior in a specific household. The model introduced in this study uses actual consumption data from 12 households over a one-year period; however, it was at a very early stage at the time, and it needed additional experiments to assess its robustness.

Conde-Clemente et al. [43]

This work builds upon the one by Trivino and Sanchez-Valdes [42], and it expands it with the definition of the LDCP (Linguistic Descriptions for Complex Phenomena) framework. The generation model relies on a corpus of sentences that are useful for mapping linguistic expressions with the corresponding range of numerical values. The framework is applied on a real-world use case using consumption data obtained within a EU-funded project (NatConsumers, now ended (https://cordis.europa.eu/project/id/65 7672, accessed on 6 December 2023); however, the use case presented in the paper is mainly for demonstration purposes. This suggests that the framework may not be fully mature for actual deployment in practical scenarios. Furthermore, as also acknowledged by the authors, the corpus of linguistic expressions may not be diverse enough, thus implying that some of the limitations of the system might have been mitigated with the addition of more linguistic data in the corpus.

Fontecha et al. [44]

The paper describes the creation of a virtual assistant, called GreenMoCa, that provides a fine-grained feedback on energy consumption and performs appliance switch operations upon user request. The system architecture consists of three main components: the conversational interface, an API service facilitating communication between the interface and the storage module responsible for saving the consumption data, and a separate API service enabling communication between the storage module and the connected smart plugs. The system underwent evaluation in a lab environment, with the participation of 10 individuals in the experiments. Nevertheless, it appears that it has not been tested in real-life scenarios.

Gamage et al. (a) [45]

In this work, the authors propose Cooee, a conversational interface to query consumption data over the energy management platform of La Trobe University, Australia. The back-end architecture of the interface comprises, besides the typical modules of a conversational agent, a pattern-matching-based module to retrieve a corresponding template of the given user's request and a Text-to-SQL component that converts the identified template into a SQL query over the database of the energy platform. The interface thus displays the result of the query along with a textual explanation of the interpreted SQL query. From a preliminary evaluation, the Text-to-SQL module appears to offer promising results compared to other state-of-the-art models; however, for more effective dialogue management, the system could benefit from further integration with other natural language processing techniques.

Gamage et al. (b) [46]

This is a follow-up work from Gamage et al. (a) [45], where Cooee's functionalities are augmented with ChatGPT (https://chat.openai.com, accessed on 30 November 2023) capabilities, precisely to address some of the main limitations encountered in the previous experiments. The addressed limitations concern, in particular, the use of a conversational context to tackle ambiguities, missing information in the user's request, or for co-reference resolution. The system did not undergo a systematic evaluation, but an exploratory analysis of the chatbot's outputs showed an adequate handling overall of these limitations. Additional tasks were also explored as integration into the Cooee system, benefiting from ChatGPT's capabilities. Among these were the analysis of tabular data, the extraction of data from unstructured sources, and the generation of recommendations. This preliminary investigation revealed the importance of domain-specific knowledge in generating

appropriate responses and avoiding the well-known hallucination problems common in generative models. Thus, the authors emphasize the need for appropriate prompt engineering strategies and domain-specific heuristics to ensure more effective integration of ChatGPT into their system.

### Giudici et al. (a) [47]

This work does not describe a conversational agent in itself, but rather the development of a framework, named CANDY, to support the design of a sustainability oriented chatbot for the home environment. The framework definition resulted from the findings of a focus group comprised of experts from multiple domains (mainly computer science, energy sector, and psychology) and was aimed at eliciting, through an iterative discussion, the main dimensions to be considered in the design of a conversational agent, among which were the possible areas of intervention, the use cases, and the possible interaction methods, as well as engagement mechanisms. The generated framework could function as a valuable and practical foundation for developers. Nevertheless, it requires additional testing and validation.

### Giudici et al. (b) [48]

The paper introduces Leafy, an app developed to encourage eco-friendly practices in households by integrating conversational agents into a smart-home setting. Specifically, the app integrates a multi-modal user–agent interaction—through a smart mirror—with gamification techniques to enhance the user engagement and their commitment to more sustainable behaviors in the domestic environment. However, the work, as described in the paper, lacks a proper evaluation of the system's usability, and the authors explicitly recognize the need for additional exploration into the potential impact of conversational interactions on improving long-term engagement.

### Gnewuch et al. [49]

This paper describes the creation of a chatbot prototype aimed at providing real-time and personalized energy consumption information. The interface design is based on four principles inspired by Design Science Research [50], and the prototype was built around two main scenarios specifically aimed at evaluating such principles. The evaluation results suggest that conversational agents are indeed a promising technology for providing energy feedback. However, the discussion with the experts highlighted the need to further explore the use of multi-modal, instead of text-only, agents. In parallel, the authors also recognize the importance of complementing the experts' perspective with focus group sessions involving lay-users in order to gain a more diverse viewpoint on the design principles devised in this work. Finally, the study presented a simulated mock-up, but an actual implementation of such principles and the deployment of the developed agent might be beneficial for a thorough validation.

### He et al. [51]

This research focuses on the study of the potential nudging effect achieved through the use of proactive virtual assistants for energy saving advice. The study involved simulations of interactions between users and a virtual assistant and showed that a significant number of users who initially provided neutral or mildly negative responses to the virtual assistant's advice were more inclined to accept the advice after a follow-up interaction with the agent. Similarly to Gnewuch et al. [49], the authors show the potentially positive impact of conversational interactions on promoting energy-saving behaviors. However, to further validate these findings a use case with a fully operational agent might be necessary.

### Hulsmann et al. [52]

The paper describes the development of a chatbot that provides information in English regarding the operation of the German energy system, starting from a model that schemati-

cally describes its functioning. The chatbot's main end users are non-expert audience and policy-makers that need this type of data to make informed decisions. The conversational interface enables them to ask general questions about energy as well as more specific questions about the energy model it is connected to (a simplified model of the German energy system is used). Users can also alter the model with their input and compare the results obtained with both original and modified models. As mentioned above, the main strength of the agent consists of making a complex engineering model more accessible to non-technical users; on the other hand, the chatbot functionalities are closely tied to the characteristics and parameters of the particular energy system model it is designed for, and further adaptations would be necessary with a different energy model.

### Ketsmur et al. [53]

The authors describe a proof of concept designed to monitor the consumption and state of home appliances and send commands to switch them on and off. In addition to the typical modules of a dialogue system, an ontology is developed to describe the data concerning appliances' consumption, the home environmental information, and the relation between the inhabitants and the appliances. A dedicated module is finally in charge of querying the ontology and retrieving the required information, building proper SPARQL queries. The system design also emphasizes aspects related to accessibility issues and restriction of actions, or if they are deemed inappropriate for certain users, such as young children. However, extensive evaluations from users are necessary to assess such design principles, particularly in relation to the accessibility issues.

### Kim et al. [24]

The authors introduce a web application that includes both a visual (a wall-mounted tablet) and a voice (using Alexa) interface that provide feedback aimed at helping users make better use of the thermostat, while at the same time promoting community-wide energy conservation through gamification mechanisms. The feedback mechanism is based on mathematical models that define (1) an energy conservation behavior score that quantifies users' efforts in reducing energy use and (2) an algorithm for personalized action recommendations along with their potential impacts. In an extensive user study, participants reported the positive influence of the personalized feedback in improving their thermostat settings. On the other hand, throughout the experiments the user engagement with the voice assistant was reportedly less frequent compared to the visual interface, thus revealing potential limitations in the conversational agent's design.

### Santos Fialho et al. [54]

The paper introduces PowerShare VA, a virtual assistant aimed at informing users about their energy usage in different forms, both with a voice-based application and through a web interface. The tool has been designed to provide consumption feedback on multiple temporal time frames (i.e., with a daily, weekly, and monthly dashboard). Real-world experiments with users in different scenarios demonstrated the assistant's capabilities in effectively interacting with users and providing meaningful feedback. On the other hand, only one participant per scenario was involved in the experiments, thus affecting the generalizability of the findings. Furthermore, the final discussion with participants elicited the need to build more user-tailored functionalities.

### Suresan et al. [55]

The authors create a chatbot that communicates the household energy consumption (e.g., concerning general trends or consumption peaks). An additional mail-alerting module is also integrated into the system to inform the user whenever consumption exceeds a given threshold. The chatbot uses consumption data obtained from an openly accessible dataset. The dataset comprises energy consumption readings for a sample of 5567 London households participating in the UK Power Networks-led Low Carbon London project

between November 2011 and February 2014. However, although the paper mentions a comparison of chatbot performance, there is a lack of informative details about how this comparison was actually conducted.

## 5. Discussion

In this section, we characterize the works outlined above along a number of orthogonal dimensions that reflect the four main research questions we defined for this review. Specifically, we aim to explore the primary research areas that deal with this particular topic, the typical use cases or scenarios in which they are employed or studied, the development methodologies applied, and, if existing, the most common evaluation practices within this domain.

### 5.1. RQ1: Research Perspectives and Primary Focus

As pointed out in the previous sections, the study of conversational agents operating in the energy sector highlights the multidisciplinary aspect of this field and the need to approach their study from different perspectives. The articles identified in this survey fully reflect this richness of perspectives, demonstrating how these types of agents are the subject of interest of different yet certainly complementary disciplines. Indeed, in an attempt to explore this aspect, we first classified the contributions collected for this survey with the relevant research areas (see Table 1) (it is worth pointing out that several papers actually included contributions from multiple fields). From this categorization, it emerged that the areas mainly explored in these papers are, as expected, NLP, HCI, and IoT/smart technologies, while, interestingly, one paper also includes a planning task. In the former case, the papers identified in this area include a description of the approaches and techniques adopted to develop the modules dedicated to managing the interaction (whether voice- or text-based). With respect to this field, two works in particular focus specifically on the generation task, with the development of a data-to-text generation framework (i.e., [42,43]), while all other works that fall into this category describe a conversational agent in its entirety, thus including a more or less detailed description of the other NLP core modules (also shown in Figure 1), i.e., those pertaining to the intent recognition step [24,41,45,46,55], the definition of allowed dialogue paths [41,48], and the generation of the agent's response [45].

The second cluster of works exhibits some commonalities from an HCI perspective, in that they tackle—either as the main focus or as an integral part of the whole contribution—the adoption of design principles or feedback strategies aimed at encouraging sustainable behaviors through the enhancement of the user engagement and experience. This goal is pursued in multiple ways, such as exploring the integration of natural-language interactions with gamification techniques, also emphasizing the importance of multimodality [24,48], or rather experimenting with actual users to define more effective design principles and frameworks [47,49]. The focus on the user also emerges through the particular attention drawn to participants' consumption profiles [54] or individual characteristics, values, and beliefs [51] while investigating users' acceptance of the eco-feedback, but also underlining the importance of properly addressing accessibility issues [53].

Fewer contributions explicitly mention the integration of IoT artifacts, such as connected devices [41,44], smart mirrors [48], and smart metering infrastructures [54], to collect and share data on energy consumption. Finally, one contribution also resorts to AI-based planning methods in order to develop a virtual assistant able to autonomously plan and execute a series of actions to address Demand Response requests (Demand Response is a flexible energy management paradigm consisting of reducing or shifting energy consumption to different times of the day in response to grid requirements, usually receiving some form of compensation in return (e.g., in terms of financial incentives—such as rewards or discounts in the energy bill—or dynamic energy pricing)).

**Table 1.** Summary of the research fields covered by the reviewed contributions.

| Research Field | Reference |
| --- | --- |
| NLP | [24,41–43,45,46,48,52,53,55] |
| HCI | [24,47–49,51,53,54] |
| IoT | [41,44,48,54] |
| Planning | [41] |

As for the type of contribution presented in each paper, a tentative classification is shown in Table 2. The vast majority of the articles collected for this survey precisely focus on describing a conversational agent, be it a chatbot or a virtual assistant. Within these papers, the content may shift either towards architectural aspects or design choices, in line with what was stated above about the different research perspectives involved, but the goal remains to introduce the agent and its possible functionalities. We deem it relevant to clarify this point in light of the selection criteria adopted for the survey and demonstrated in Section 3. The fact that papers not necessarily focused on describing this type of conversational agent, but concerning closely related aspects, were also considered eligible, allowed for the inclusion of other types of contributions. As a result, and also as anticipated above, we identified two studies devoted to the development of rule-based NLG techniques aimed at generating textual eco-feedback from consumption data [42,43]. The work of Giudici et al. [47], on the other hand, proposes a framework to support the development of conversational agents geared towards energy sustainability in the home; the framework includes a number of dimensions to be taken into account in the agent's design. Finally, the article by He et al. [51] explores the impact of proactive virtual assistants on occupants' energy-saving behaviors related to HVAC operations, also highlighting the significance of the individual characteristics on participants' responses.

**Table 2.** Summary of the main research focus of the reviewed contributions.

| Main Focus | Reference |
| --- | --- |
| Description of the agent | [24,41,44–46,48,52–55] |
| Data-to-text architecture description | [42,43] |
| Behavioral study | [51] |
| Framework definition | [47] |

*5.2. RQ2: Main Goals and Use Scenarios*

This section is devoted to an overview of the goals, in terms of type of feedback and, more in general, of the functionalities provided in the systems included in this survey (also summarized in Table 3). At the same time, we intend to identify the main use cases and scenarios—either envisioned or concretely experimented—adopted in the surveyed studies (see also Table 4). This is in order to highlight the diversity of objectives among the agents explored in this area, while also defining their typical operational scope.

Among the systems identified in our collection, some focus on monitoring consumption and collecting and analyzing data related to users' main habits. This type of approach is often geared towards presenting detailed information about a given device or service usage, providing the user with useful and comprehensible insights into their consumption behaviors, even resorting to different time resolutions (e.g., daily, weekly, monthly). On the other hand, eco-feedback systems can also be designed with the aim of offering more practical suggestions to users. The latter, however, mainly deal with general advice concerning good practices and recommended behaviors; more targeted and personalized recommendations based on user-specific data and context are provided in MySmartE [24], where the feedback comes as a result of a mathematical model that computes an energy score for each user, and in the NLG architecture described in Trivino and Sanchez-Valdes [42] and Conde-Clemente et al. [43], where different templates were used based on different user models. It

is worth noting that conversational agents, and the ones reviewed here in particular, can be designed with very different goals. As a result, in many cases monitoring and proactive feedback functionalities are not mutually exclusive, but rather go hand in hand; likewise, they can be integrated with other types of functionality such as those related, especially in smart-home environments, to the automation of commands on smart devices.

**Table 3.** Summary of the main goals and functionalities of the systems described in the reviewed contributions.

| Main Goal(s) | Reference |
|---|---|
| Consumption monitoring | [24,42–46,48,49,53–55] |
| Energy-saving advice | [42–44,48,51,54] |
| User-tailored | [24,42,43] |
| Smart-home automation | [24,44,53] |
| Other | [41,47,52] |

As for the analysis of the main usage scenarios in which these systems are developed or designed, a relevant aspect that emerges is that, in the vast majority of the cases considered, the main application pertains to the domestic sphere. This seems to imply that such systems are conceived primarily to meet the needs of household users. However, we pinpointed one particular case in which experimentation extended beyond the home environment: in the project by Santos-Fialho et al. [54], the virtual assistant was also implemented in an industrial facility and in a restaurant in order to show the adaptability of the conversational agent to heterogeneous scenarios. In addition, some case studies move away from the conception of specific physical environments to rather focus on energy data processing models and platforms. A significant example is the project presented in Gamage et al. [45] (and in their follow-up work [46]), where the chatbot they developed was integrated within the Latrobe Energy Analytics Platform (LEAP), a platform built for the energy management and optimization of net zero carbon emissions at La Trobe University, Australia. Similarly, in the research conducted by Hulsmann et al. [52] the data used to generate the feedback was retrieved from a model that simulates the operation of Germany's energy system, enabling users without technical knowledge of energy system modeling to effortlessly retrieve simulation output information, modify model settings, and then launch new simulations.

**Table 4.** Context of use (either envisaged or actually implemented) of the systems described in the reviewed contributions.

| Context of Use | Reference |
|---|---|
| Household | [24,41–44,47–49,51,53–55] |
| University campus | [45,46] |
| Other | [52,54] |

### 5.3. RQ3: Systems' Development

This section of the survey is devoted to aspects more closely related to the development of conversational agents and systems for text-based energy feedback. The research question guiding this part focuses on the approaches adopted in such systems, analyzing the use of available platforms and the implementation choices (see the summary in Table 5), the intended modes of communication (Table 6), and the language employed. It is worth pointing out that in this section the focus falls specifically on the reviewed contributions where a system has actually been developed (whether in the form of a preview or of a fully functional agent).

Concerning the implementation choices, we observe that in a few cases the whole system has been developed without resorting to any external framework or library [42,43].

Giudici et al. [48] combined a rule-based implementation of the dialogue manager (using a finite state machine) with external libraries, respectively, for the NLU engine (i.e., NLP.js (https://github.com/axa-group/nlp.js/, accessed on 30 November 2023) and for the speech modules (using DeepSpeech (https://github.com/mozilla/DeepSpeech, accessed on 30 November 2023)). Gamage et al. [45] fine-tuned a pre-trained BERT model [56] for the entity recognition task, thus resorting to the HuggingFace Transformers library [57]; they also used the Dateutil parser library in the Text-to-SQL module to convert temporal expressions in natural language to standard SQL DateTime format. In their further development of Cooee [46], they finally resorted to ChatGPT to improve some of the NLU and DM tasks. In the remaining cases, the related authors relied either partially or entirely on pre-existing development platforms available online (some of them were also mentioned in Section 2.2). For three agents, the preferred platform was RASA (see Table 5), but in particular in Alexiadis et al. [41], the built-in modules available in the platform for NLU and dialogue management were customized to properly handle multi-intent requests from the users and to initiate the relevant planning actions while tackling a Demand Response request prompted by the user. On top of that, a speech module is added to enable voice interactions. Two other personal assistants were built, relying on the services provided by Google, i.e., DialogFlow [44] and Google Actions [54]. Gnewuch et al. [49] created an interactive prototype of their agent using BotPreview (https://www.botpreview.com/, accessed on 30 November 2023), a platform for building previews of chatbot interactions that, however, has been shut down since April 2020. To develop their voice-based assistant, Kim et al. [24] relied instead on the Alexa Skill Kit (https://www.developer.amazon.com/alexa/alexa-skills-kit, accessed on 30 November 2023), the toolkit provided by Amazon to create a "skill", i.e., a custom application that can be used through an Amazon Echo device and with Alexa as the conversational interface. Finally, Ketsmur et al. [53] used IBM Watson, adapting its NLU module to properly recognize entities referring to the involved device, the corresponding home partition, and the type of energy resource (i.e., electricity, water and gas), as well as to identify a wider range of date formats.

**Table 5.** Summary of the implementation frameworks adopted by the systems described in the reviewed contributions.

| System Implementation | Reference |
| --- | --- |
| Own implementation | [42,43] |
| With ext. libraries | [45,46,48] |
| RASA | [41,52,55] |
| Google services | [44,54] |
| Other frameworks | [24,49,53] |

The investigation into the preferred modes of communication adopted by these agents aims to complement the analysis of the development approaches, in that it draws particular attention to the favored interaction mechanisms. We point out that this aspect refers only to the modes used to interact with the conversational agents, thus excluding possible additional applications, such as web interfaces or gaming functionalities, integrated into the whole system, as in MySmartE [24] or Leafy [48]. The distribution of the interaction modalities, also reported in Table 6, shows that text-based communication (i.e., using chatbots) is the choice more extensively studied in the development of these agents. The adoption of voice-based assistants falls behind, while multi-modality (which enables both) is only explored in three works.

**Table 6.** Summary of the main modes of interaction adopted by the systems described in the reviewed contributions.

| Modality | Reference |
| --- | --- |
| Text | [42,43,45,46,49,52,55] |
| Voice | [24,41,44,51] |
| Multimodal | [48,53,54] |

Finally, the role of language in interactions with agents of this type is another aspect that we believe is crucial in our analysis. Indeed, we observe that the vast majority of them adopted English as the main language, despite the fact that some systems were developed and tested in non-English-speaking countries, such as Greece [41] and Germany [49,52]. The only few exceptions are made by the GreenMoCa assistant [44] and the proof of concept presented in Ketsmur et al. [53], which were developed for Spanish and European Portuguese, respectively. In most of the remaining articles, the language used was not explicitly mentioned and the relevant information was derived primarily from the graphical examples (e.g., screenshots) proposed by the respective authors. It is worth clarify, however, that it was often unclear whether the examples were the result of direct translations of the original interactions or whether English was actually set as the system's default language.

*5.4. RQ4: Evaluation and Common Practices*

In this section, we complete the discussion part with a comparison of the evaluation approaches adopted for the systems examined in this survey. Being aware of the challenges inherent in the evaluation of conversational agents, regardless of their type, we sought to organize the relevant information retrieved from the articles in the collection (see Table 7). Our goal is to identify shared approaches or, conversely, to highlight the lack thereof.

A preliminary remark concerns the fact that, in many cases, the system presented was not subjected to any form of evaluation, showing once again the difficulties that can be encountered in establishing the actual functioning and effectiveness of this type of applications. Among the works, on the other hand, where some form of evaluation was applied, we observe that in a good number of cases this was done with the involvement of human participants, highlighting the central role of the user for the actual deployment of these tools. Human involvement, in turn, is reflected through different common practices, which typically include interviews or questionnaires: in Alexiadis et al. [41], participants were requested to perform a series of actions while interacting with the agent, and eventually indicate the success or failure of those actions. They would then answer questions about the usefulness and acceptance of the agent. A similar approach is followed in Fontecha et al. [44], where participants were asked through a questionnaire to evaluate the usability of the agent and the user experience in relation to the actions performed by the agent. A common approach is also shared in Kim et al. [24] and Santos-Fialho et al. [54], in that, in both cases, users were required to install all the necessary devices, enabling the evaluation of the agent in real-world contexts; in addition, they both conducted post-experiment inquiries to assess the systems' usability and the usefulness of the provided feedback. To summarize, the main criteria users were required to evaluate in this setting include the following:

- correctness of the requested action [41,44];
- easiness of communication [24,41,55];
- frequency of the interactions [24,41,54];
- usefulness of the feedback [24,44,54].

Given the design-oriented focus in Gnewuch et al. [49], instead, a focus group was established, involving experts in the field; the evaluation included a general discussion on the quality of the chatbot's responses and a more detailed examination of each design principle.

Alternatively, qualitative methods were adopted in a consistent portion of the surveyed works; the ones analyzed here range from purely descriptive remarks on the resulting

conversations [46,53], on the generated messages [42,43], or on the dialogue flow [41], to more complex considerations derived from a systematic analysis of consumption before and after system adoption [24].

Finally, a smaller percentage of system descriptions also included an evaluation step using typical text classification metrics. In an attempt to also provide an assessment of the agent's ability to adequately recognize user requests, in both Hulsmann et al. [52] and Alexiadis et al. [41] the intent recognition module was evaluated in terms of F-score (which in Alexiadis et al. was micro-averaged). Finally, in Cooee [45], the Text-to-SQL module (developed to translate users' requests into SQL queries on the LEAP databases) was evaluated, comparing its accuracy with two state-of-the-art methods, i.e., tabular Question Answering and sequence-to-sequence classification, both implemented using models and datasets from the HuggingFace library.

**Table 7.** Summary of the main categories of evaluation approaches (when applied) described in the reviewed contributions.

| Evaluation | Reference |
|---|---|
| User study | [24,41,44,49,54,55] |
| Qualitative analysis | [24,41–43,46,53] |
| Classification metrics | [41,45,52] |

### 5.5. Main Findings and Implications for the Field

In this closing part of the discussion, we aim to summarize the main findings we can draw from the comparisons carried out in this section, in order to answer the four research questions formulated at the beginning of Section 3 and identify the implications they may have on the future design of conversational agents for energy feedback.

To answer RQ1 (Section 5.1), it becomes clear from this preliminary analysis that there is an interest in conversational agents that provide some form of eco-feedback, but what stands out more clearly is the need to take an interdisciplinary approach in the study of these agents to fully understand their implications and maximize their positive impact in the energy sector. In fact, it is important to take into account the specifications and needs expressed both by experts and target end users, so as to devise more effective forms of eco-feedback. This is also reflected in the research focus of the papers included in this survey, which sometimes deviate from the pure description of the agent and broaden the scope of the study to precisely emphasize these latter aspects.

As for RQ2 (Section 5.2), the systems reviewed in this survey exhibit a certain degree of versatility in terms of feedback goals, often combining monitoring and proactive feedback functionalities. These functionalities, however, are less often integrated with automation features, thus somehow limiting the operational scope of the agent, especially when it comes to efficiently handling energy resources and devices. Furthermore, in terms of usage scenarios, the systems considered show a greater preference for domestic applications; nonetheless, the remaining use cases analyzed suggest that these agents have considerable potential for use in both commercial and industrial settings, but also as user-friendly consultation interfaces for large-scale energy management systems.

To consolidate the observations that emerged from RQ3 (Section 5.3), we notice a great diversity of implementation choices that also include the combination of rule-based and data-driven approaches, depending on the task at hand: NLU tasks concerning intent detection or recognition of specific entities (e.g., devices or settings) are typically based on custom training datasets, while dialogue managers are often built using hand-crafted rules attempting to include all the possible interaction paths. Furthermore, text-based systems have raised a greater research interest so far, although the widespread adoption of personal assistants for a wide range of daily tasks may suggest a shift in the future towards multi-modal settings, even for this type of agent. Language-wise, English seems to stand out as the preferred language for the development of these agents, although, most notably,

this type of information is often elicited from dialogue excerpts provided in the relative papers, and not explicitly declared by the authors. A clear statement on this matter is thus essential and always advisable [58].

To finally answer our last research question (RQ4, Section 5.4), the existing body of work on conversational agents for energy feedback does indeed involve evaluation in some cases, with common practices including human participant involvement (mostly through interviews and questionnaires). Forms of automatic evaluations are not reported, except for specific sub-tasks such as intent recognition or Text-to-SQL generation. At the same time, we observe the absence of a systematic approach in the evaluation methodology, which in some cases is limited to purely qualitative observations or, alternatively, is not contemplated at all. Such heterogeneity underscores the need for a greater consistency in the evaluation practices.

## 6. Open Challenges and Possible Future Directions

Building on the observations reported in the previous section, this concluding part of the survey puts forth some reflections on the envisioned research directions concerning system design, evaluation, and actual deployment, also taking into account the technical as well as the social challenges these agents may have to address in the near future.

### 6.1. Goals and Usage Scenarios

In terms of usage scenarios, we have reported that the vast majority of contributions on this topic focus on domestic settings, and in general on single utilities [54]. However, we foresee some major extensions in the landscape of the possible feedback goals and use cases of these agents.

The push to exploit resources from renewable sources has paved the way for an innovative economic phenomenon: prosumerism. The related term "prosumer" comes from the combination of the words "producer" and "consumer" and refers to individuals or entities who, through access to alternative energy sources to the main grid, can not only meet their own energy needs but also generate surplus energy to share with the grid as well as with other users. This evolution introduces a new paradigm in eco-feedback design, requiring a shift from the traditional consumer-oriented approach to one more focused on the needs of this new energy actor; prosumer-oriented feedback should thus provide useful insights into how best to manage their energy balance and their relationship with the main grid [59].

This transition also brings with it important social implications. Namely, due to the possibility of sharing their surplus energy with other users, prosumers can become part of decentralized energy networks, giving rise to Renewable Energy Communities (RECs), i.e., groups of people or entities that produce, consume, and share energy, mostly renewable, within a limited geographic area. In this emerging reality, we identify a significant area where conversational agents for energy feedback can be effectively applied. In fact, the potential exerted by eco-feedback within a REC setting has already been investigated [22,60]. Among the works analyzed for this survey, MySmartE [24] currently constitutes the only one that approaches this direction. However, it is important to note that, in their contribution, the focus on this aspect results in the adoption of gamification techniques, thus not contemplating the use of the voice interface for the provision of energy feedback in community settings. Nonetheless, we believe that the expansion of conversational agents to energy communities can be useful for monitoring purposes, thus providing real-time reports on consumption trends at both individual and collective levels, but most importantly as a tool of pro-active feedback. For instance, a recent work by Jensen et al. [60] proposed the "Community Energy Planner", an app that enabled both individual and collective views of energy feedback (albeit without relying on conversational interactions). Feedback at the collective level in particular allowed members to monitor the consumption patterns of the community over time. The app was also able to provide insights into total savings in kWh and euros, predictions of the energy consumption for the day within the community,

and the extent to which the group had collectively adhered to the recommendations. While it is crucial to carefully design the feedback in order to avoid potential downsides, such as users feeling guilty when goals were not achieved [60], this prospect can assist individuals in aligning their consumption habits with community requirements, trying to find a middle ground between their personal needs and the collective dynamics.

*6.2. System Design*

Within the landscape of features and modules integrated into the conversational architecture, the observations that emerged in Section 5 outline two key aspects, which are also strictly related to the above-mentioned RECs scenario. First, with the only exception of Santos-Fialho et al. [54], there is a lack of attention among the surveyed works to monitor the consumption and production of energy from renewable sources, produced, for example, through the use of photovoltaic panels or wind turbines. Their increasing spread and greater accessibility by an ever-widening audience of consumers underlines the desirability of extending the capabilities of conversational agents to also provide feedback on energy production from these sources and on users' self-consumption. The same applies to Battery Energy Storage Systems (BESSs), which further enhance the benefits coming from renewable energy by allowing its storage during overproduction, and its release during times of low or no production. Similar to other monitoring platforms based on purely visual techniques, the feedback provided in this context could ideally include detailed monitoring of the PV system performance, with both real-time and historical data on solar energy production. Furthermore, it might offer the possibility to examine crucial aspects, such as energy flows to and from the different energy sources involved, i.e., the main grid, the PV system, and, if present, the battery storage system. The agent might also provide reports on the self-consumption and self-sufficiency of the system. Within the context of renewable energy systems, self-consumption (SC) is defined as "the amount of electricity locally generated and consumed with respect to the total local generation" while self-sufficiency (SS) is defined as "the consumption amount supplied by local generation with respect to the total consumption" [61] (p. 2).

The second crucial aspect focuses on the need to develop advanced data processing modules, devoted, for example, to optimization, planning or forecasting tasks. In fact, with the expression "advanced data processing," we broadly refer to any computational task that goes beyond the retrieval of consumption data from external databases or smart metering infrastructures. Surprisingly, only two of the studies included in our analysis explicitly address this issue. Alexiadis et al. [41] use planning techniques to intervene on HVAC and lights and adjust their power consumption, e.g., by changing the fan speed or dimming lights at a given percentage, taking into account additional factors such as the human presence in a given room and the outdoor temperature. In MySmartE [24], instead, a mathematical model is defined to quantify users' efforts in reducing energy usage based on the household's time series of thermostat consumption. Besides the studies analyzed in this survey, further examples in this direction might include the integration of modules that implement classical optimization techniques, such as multi-objective Mixed Integer Linear Programming (MILP). This method involves formulating and solving optimization problems with multiple objectives and constraints, making it suitable for scenarios where simultaneous optimization of multiple goals is required, such as in the energy communities also mentioned above. MILP approaches have proved useful when attempting to find optimal solutions with competing goals, such as economic cost, environmental impact, and social components [62]. Optimization algorithms can also be incorporated to compute a more efficient battery usage, finding optimal charging and discharging times, or to schedule the use of household appliances [63]. Time series models can finally be leveraged to analyze historical energy consumption data and predict future energy needs [64]. For the enhancement of energy efficiency, we believe that moving forward in this direction is essential to develop conversational assistants that can proactively improve energy efficiency and provide effective and personalized energy-saving feedback to users.

In addition, we have stressed in this paper the centrality of the user in the development of these dialogue agents. It is important to recognize in this regard that users in fact may have different motivations and interests while using such tools. Some may be highly motivated by their sensitivity to climate change issues, while others may be driven primarily by economic motivations (typically savings in their utility bills). They may also have a varying degree of energy literacy, and of familiarity with smart tools and user interfaces. In addition, they may have a greater or lesser inclination to conform to energy optimization practices while sacrificing their personal comfort. In some of the studies identified in this survey [24,41,51], variables of this type were considered during the evaluation phase to assess their possible correlation with the agent's usability and acceptance. Those studies indeed showed a significant correlation between the users' positive assessments overall of these agents and their opinions regarding environmental issues, as well as their familiarity with these tools or their thermal preferences. We believe that in order to develop a more persuasive conversational agent, it is critical to consider these factors, even in the early stages. User modeling, for example, is a common stage in natural language generation techniques: it consists of collecting and processing users' preferences, characteristics, and needs in order to customize the generated text. Several works can be found in the literature on the adoption of user modeling approaches [65,66], especially to build more persuasive dialogue agents in specific domains, such as diet coaching [67,68]. We believe that, even within the energy domain, the availability of information of this type can be of help. Among the surveyed works, Trivino and Sanchez-Valdes [42] resorted to a user model to create user clusters based on their consumption habits. Additional examples of adaptive feedback based on user modeling may include different levels of technical detail present in the text according to the user's prior background, or a shift in the focus of energy efficiency in terms of emissions saved or possible savings on the bill, based on the user's motivations. This could be a viable approach to more effectively engage a wider and more diverse audience of target users.

Finally, as remarked in Section 5, English is by far the preferred language adopted by the reviewed systems to provide the eco-feedback. Although English is widely recognized as the vehicular language, its actual deployment in real-life scenarios may have a significant impact on the applicability of this type of conversational agents, even more so if deployed in residential contexts with different end-user demographics. In general, the use of a non-native language to interact with the agent poses an important challenge to accessibility and to the acceptance of the agent itself, ultimately reducing its effectiveness as an actual means of behavioral change. We therefore emphasize the importance of creating conversational agents for energy feedback that cater to diverse language backgrounds, ensuring a greater accessibility and inclusivity, thus further enhancing the potential impact of these agents on users' engagement and long-term adoption of more sustainable behaviors.

### 6.3. Evaluation

The significant disparity in the approaches and objectives of each system, as highlighted in Section 5, makes any meaningful comparison nearly unfeasible. Nonetheless, a greater alignment in evaluation practices would be desirable in the near future. To this end, we suggest as a further development the creation of a comprehensive evaluation framework, in line with similar research works proposed both for open-domain conversational agents [69,70] and conversational agents for smart-home environments [71], to offer a clearer guidance and support for researchers and developers in the assessment of both feedback and dialogues' success.

In addition, and drawing inspiration from what was also proposed in Kim et al. [24], further research might be encouraged to assess the actual impact of the use of such agents on long-term behaviors, investigating, for example, whether and to what extent energy saving could be effectively attributed to user engagement through dialogue and resorting to text-based proactive feedback. In fact, this approach has been adopted in the past in relation to other types of energy feedback (see Zanghieri et al. [72] for a meta-analysis of

such works); its extension to conversation-oriented eco-feedback applications would have the twofold benefit of enabling a more exhaustive understanding of their potential, both for their own sake and also compared to alternative and more common strategies.

## 7. Conclusions

In this article, we provided an overview of the literature regarding conversational agents and feedback generation in the energy sector. We primarily remarked that there is a vast amount of research on providing feedback through visualizations or using gamification techniques to influence user behaviors. In contrast, work that focuses on user engagement through conversational tools and interfaces is more limited. We attribute this to the very specific nature of these conversational agents on the one hand, but also to the selection criteria we decided to follow on the other. Concerning the latter in particular, we adopted very strict criteria, precisely in order to focus on work that encompassed both of the two key points that were in line with our research focus, i.e., the delivery of eco-feedback through dialogue-based interactions; this greatly narrowed the range of eligible works. Although we cannot exclude that the various filtering and screening steps were influenced by unintentional biases, we believe that, given the specialized nature of our research topic, the selected number of articles is still representative and it adequately captures the diversity of this type of conversational agent.

In addition to the methodological considerations, there are several implications of our findings for this field that should be considered, as outlined in Section 5.5. First, we have highlighted the importance of taking an inter-disciplinary approach in the development of conversational agents for eco-feedback. This translates, for example, into a greater centrality given to the user in both the design and evaluation phases of these agents. With regard to the potential capabilities of the agents at hand, a further implication could involve greater synergy than that evidenced in the reviewed papers between eco-feedback and automation capabilities. This aspect aims to expand the agents' operational capabilities when it comes to a more efficient management of energy resources and devices. Finally, the diversity of usage scenarios, currently oriented predominantly towards domestic applications, suggests that developers might explore and design conversational agents with features suitable for broader contexts, including commercial and industrial settings. In parallel, we deem it critical that future developers consider the possibility of a shift towards multimodal settings and, more generally, towards making these tools more accessible to a wider audience (e.g., elderly or visually-impaired people). The significant advances in machine learning and NLP, combined with the growing adoption of smart devices on one side and the greater acceptance of conversational agents in everyday tasks on the other, makes the potential of these tools for energy purposes even greater.

In conclusion, despite the relatively limited contribution given to conversational agents in the energy context to date, we believe that their role will become increasingly central in supporting users in the efficient management of energy consumption, thus helping to promote greater awareness and sustainability in the use of energy resources, both individually and collectively.

**Author Contributions:** Conceptualization, M.S.; methodology, M.S.; writing—original draft preparation, M.S.; writing—review and editing, M.S. and M.A; funding acquisition, M.A. All authors have read and agreed to the published version of the manuscript.

**Funding:** We acknowledge financial support under the National Recovery and Resilience Plan (NRRP), Mission 4 Component 2 Investment 1.5—Call for tender No. 3277, published on 30 December 2021 by the Italian Ministry of University and Research (MUR), funded by the European Union-NextGenerationEU. Project Code ECS0000038—Project Title eINS Ecosystem of Innovation for Next Generation Sardinia—CUP F53C22000430001-Grant Assignment Decree No. 1056, adopted on 23 June 2022 by the Italian Ministry of University and Research (MUR). This work was also partially funded under the National Recovery and Resilience Plan (NRRP), Mission 4 Component 2 Investment 1.3—Call for tender No. 1561 of 11.10.2022 of the Italian Ministry of University and Research (MUR), funded by the European Union–NextGenerationEU, Project code PE0000021, Concession Decree

No. 1561 of 11.10.2022, adopted by the Italian Ministry of University and Research (MUR), CUP F53C22000770007, according to attachment E of Decree No. 1561/2022, Project title "Network 4 Energy Sustainable Transition–NEST". We further acknowledge the support by Fondazione di Sardegna-UniCA, through grant CUP F75F21001220007 for project ASTRID Advanced learning STRategies for high-dimensional and Imbalanced Data.

**Data Availability Statement:** Not applicable.

**Conflicts of Interest:** The authors declare no conflicts of interest.

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
