# Peer review of "Conversational Agents for Energy Awareness and Efficiency: A Survey"

_electronics, doi:10.3390/electronics13020401_

Round 1

Reviewer 1 Report

Comments and Suggestions for Authors

The paper presents a survey on the use of conversational agents for energy awareness and efficiency. The authors emphasize the significance of reducing greenhouse gas emissions and promoting energy efficiency. They discuss how conversational agents can play a crucial role in providing users with personalized advice and detailed information to enhance energy efficiency. The main objective of the article is to explore the contribution of conversational agents in achieving the energy transition and sustainable development goals.

I have some comments to improve the scientific depth of the paper:

1- The authors should provide a more detailed explanation of the different sub-types of conversational agents, such as chat-oriented agents and task or goal-oriented dialogue systems.

2- It would be beneficial to provide more information on the specific techniques and algorithms used in the development of conversational agents, such as natural language understanding, dialogue management, and natural language generation.

3- The authors should explore the potential applications of conversational agents beyond providing energy-related feedback, such as in customer service or healthcare.

4- The authors should provide information on how the data from the selected articles were extracted and organized for analysis. This could include details on the specific data points collected and any coding or categorization methods used.

5- The authors should include more critical analysis of the limitations and implications of the research. This could involve discussing potential biases or shortcomings in the methodologies used, as well as considering the broader implications of the findings for the field.

6- The Systems' design subsection briefly mentions the importance of extending the capabilities of conversational agents to provide feedback on energy production from renewable sources. However, it could delve deeper into the specific ways in which these agents can integrate and provide feedback on renewable energy generation and self-consumption.

7- The authors mention the need to develop advanced data processing models but does not provide further details on what these models could entail.

8- It would be helpful to include examples or explanations of the types of data processing techniques that could be employed to enhance the functionality of conversational agents.

Comments on the Quality of English Language

Minor editing of English language required

Reviewer 2 Report

Comments and Suggestions for Authors

The manuscript presents a survey of recent advancements in conversational agents for energy awareness and efficiency. The authors conducted a series of steps to figure out the target papers for the literature review, involving searches in academic databases, filtering out unrelated works, and excluding non-peer-reviewed papers. The manuscript compares 12 articles from four dimensions: 

- Research perspectives and primary focus

- Main goals and user scenarios

- Systems’ development

- Evaluation and common practices

The comparison and findings could potentially attract further research in this domain.

A few suggestions/comments for this manuscript:

1. A few paragraphs start without an indentation, e.g. Line 135, Line 491, and Line 504. I recommend the authors proofread the paragraphs to ensure the consistency of the formats.

2. Figure 3 shows that a limited set of papers were selected for discussion in this manuscript. No publications in 2016, 2017, and 2019 were selected. This potentially indicates a limited scope, contradicting the broader scope claimed at the beginning of this manuscript. Some more clarification would be beneficial.

3. Table 2 uses “CA” as an abbreviation for conversational agent without an explicit definition in the manuscript. I suggest explicitly marking it in the manuscript for better readability.

4. The reviewer would like more clarification of the criteria used for assessing the usage of external libraries, as depicted in Table 5. Take the paper "Cooee: An Artificial Intelligence Chatbot for Complex Energy Environments" as an example, the authors listed it as not using any external libraries. But according to the project website (https://gihanmora.github.io/project_pages/cooee/), the authors state that they used BERT and other NLP techniques like NER which typically require external libraries (e.g. nltk, SpaCy, huggingface). 

5. There is a single paper for each of Alexia Skills Kit, BotPreview, and IBM Watson. This potentially indicates an inadequacy/incompleteness in the criteria used for comparison. I suggest restructuring the comparison of system implementations.

6. Line 610 - Line 613 categorizes papers based on the types of user study. I suggest merging the comparison into Table 7.

Comments on the Quality of English Language

N/A

Reviewer 3 Report

Comments and Suggestions for Authors

The paper’s subject is interesting. The organization of the paper is generally well. Some modifications are needed to improve the quality of the paper.

1-     The authors have declared that the aim of this article is thus to examine the landscape of the use of conversational agents for energy awareness and efficiency. They have tried to examine related techniques from Natural Language Processing for conversational agents for energy awareness and efficiency. It seems the main question that authors should answer is possible future directions. Hence, the paper should be modified to highlight the possible future directions.

2-     What is the gap in available research works and references? It is recommended to clarify the gaps in the literature. The references should be reviewed in detail. The advantages and disadvantages of all references should be noted clearly. Clarifying the research gaps helps further future research works.

3-     Are there similar surveys and review papers in the field of conversational agents for energy awareness and efficiency? If yes, it is recommended to illustrate the advantages of this paper compared to available ones. Also, other surveys about the conversational agents could be useful.

4-     The abstract should be revised. Some outcomes of this survey and literature review should be addressed in the abstract.

5-     Is it possible to show a framework or perspective of the reviewed field in the abstract?

6-     Addressing the most important subject in the abstract is recommended.

7-     New relevant papers and references should be cited and reviewed.

8-     Some visualizations have been used in this review paper. However, it seems that additional visualizations are needed.

9-     Table captions should be aligned at the top of the tables.

10-  A categorized summary of the reviewed studies and their features should be added to the paper. Some Tables have tried to summarize the reviewed papers. However, they should be improved and enriched by additional information and features.

11-  The unnecessary parts should be removed from the conclusion. The answers to the main question should be discussed in the conclusion. Possible future directions could be highlighted in the conclusion.

12-  The size of the contents in figures, such as Fig. 1, should be revised. The bigger size is better for writing in figures.

13-   Are 5 references for the review, as shown in Fig. 3, satisfying? Is it necessary for more references and studies to be reviewed?

Comments on the Quality of English Language

Moderate editing of English language required

Round 2

Reviewer 1 Report

Comments and Suggestions for Authors

The authors have improved the manuscript and it's ready for publication in its current form 

Reviewer 2 Report

Comments and Suggestions for Authors

The authors have addressed all my comments. I suggest accepting the manuscript.

Reviewer 3 Report

Comments and Suggestions for Authors

The authors have tried to respond to review comments.

Comments on the Quality of English Language

 Minor editing of English language required